# Significance of the Gut Microbiome for Viral Diarrheal and Extra-Intestinal Diseases

**DOI:** 10.3390/v13081601

**Published:** 2021-08-12

**Authors:** Ulrich Desselberger

**Affiliations:** Department of Medicine, University of Cambridge, Cambridge CB2 0QQ, UK; ud207@medschl.cam.ac.uk

**Keywords:** gut microbiome, microbiome–host relationship, antiviral immune responses, gut disease, noninfectious disease, microbial metabolites, microbiome transplantation, probiotics, prebiotics, diet

## Abstract

The composition of the mammalian gut microbiome is very important for the health and disease of the host. Significant correlations of particular gut microbiota with host immune responsiveness and various infectious and noninfectious host conditions, such as chronic enteric infections, type 2 diabetes, obesity, asthma, and neurological diseases, have been uncovered. Recently, research has moved on to exploring the causalities of such relationships. The metabolites of gut microbiota and those of the host are considered in a ‘holobiontic’ way. It turns out that the host’s diet is a major determinant of the composition of the gut microbiome and its metabolites. Animal models of bacterial and viral intestinal infections have been developed to explore the interrelationships of diet, gut microbiome, and health/disease phenotypes of the host. Dietary fibers can act as prebiotics, and certain bacterial species support the host’s wellbeing as probiotics. In cases of *Clostridioides difficile*-associated antibiotic-resistant chronic diarrhea, transplantation of fecal microbiomes has sometimes cured the disease. Future research will concentrate on the definition of microbial/host/diet interrelationships which will inform rationales for improving host conditions, in particular in relation to optimization of immune responses to childhood vaccines.

## 1. Introduction

The human gut microbiota, comprising bacteria, viruses, fungi, protozoa, and parasites and comprehensively termed the gut microbiome, have received increased attention for about a decade, when it was recognized that their commensal or symbiotic relationship is of great importance for human health, including immune responses correlated with protection from infection or disease [1,2,3,4]. Most of the microbiome data relate to the bacteria (bacteriome) and viruses (virome) populating the gut. Observational studies initially reported on cotemporal correlations of the composition of the gut microbiome with immune responses or disease outcome [5,6]. More recent studies aimed at identifying causal relationships between metabolic products of the gut microbiome and the host in health and disease [7,8,9]. This review emphasizes the importance of the transition from observational correlation studies to studies exploring causal microbiome–host relationships, which will provide data for rational developments of microbiota as probiotic agents.

## 2. The Intestinal Microbiome

The main bacterial phyla present in the gut are *Proteobacteria, Bacteriodetes, Firmicutes,* and *Actinobacteria*, with their total number estimated to be 10^14^ particles, and populating mainly the colon [10,11]. The gut microbiota in infants are originally similar to those of the mother but will develop by colonization with *Bifidobacterium, Bacteroides,* and *Clostridium* spp. [12], and the composition of the gut microbiome will highly depend on nutritional/feeding and environmental conditions [4,13]. Viruses found in the gut are mainly members of the *Picornaviridae, Reoviridae, Caliciviridae,* and *Astroviridae* families, and of various families of bacteriophages; in addition, members of the *Adenoviridae, Picobirnaviridae, Herpesviridae,* and *Retroviridae* families can be present [14]. Many bacteria replicating in the gut are commensals or symbionts; some bacteria, viruses, and most protozoa and parasites are pathogenic. The microbial homeostasis in healthy individuals can be disturbed to become a ‘dysbiosis’, which may be associated with the development of disease [15]. It has been extensively documented that the gut microbiota of children growing up in low- or middle-income countries differ drastically from those of children in high-income countries [6,10,13]. The composition of the gut microbiome is of great importance for the development of a functional immune system, which defends against pathogenic microbes [16].

Viruses and bacteria interact in the gut in a complex way. Thus, bacteria exhibiting cell receptor-like molecules on their surface can interact with viruses, and the complexes may either be washed out of the gut or be taken up by gut epithelia [17,18]. Accordingly, treatment of mice with antibiotics (ABs) reduced the diarrhea caused by murine rotaviruses and enhanced rotavirus-specific IgA responses [19] or reduced the uptake of poliovirus/reovirus–*Bacillus cereus* complexes [20]. Norovirus infectivity was also reduced in AB-treated mice [21,22]. Mouse microbiota can be reconstituted after AB treatment or in germ-free (GF) animals: while the AB option is inexpensive, it may not eliminate all residual bacteria and also affect epithelial cells; the GF option is more cumbersome and may be affected by developmental defects [23]. In human adults, AB-mediated microbiome changes were shown to increase the replication of rotavirus vaccine [24]. Thus, enteric viral infections can be facilitated or inhibited by bacteria, and in turn, latently virus-infected animals can become resistant to particular bacterial infections [25,26]. Particular immunodeficient mouse strains were found to be resistant to rotavirus infection and disease. By treating the microbiota of these mice with heat, filtration, and ABs, it was discovered that the resistance was due to the presence of segmented filamentous bacteria (members of *Clostridiales*) growing in the terminal ileum, which seemed to directly neutralize the virus, possibly by interfering with its binding to host cell receptors. This represents a novel way of protecting mammals against rotavirus disease [27]; the cytokines IL-22 and IL-18 were also found to be involved in this protection [28]. In RV-infected neonatal mice, a loss of *Lactobacillus* spp. was detected in the ileum on day 1 p.i., accompanied by an increase in *Bacteroides* and *Akkermania* spp., which both digest mucin glycan. Simultaneously, a loss of mucin-producing goblet cells was observed, which had recovered on day 3 p.i. [29]. These data indicate that resident bacteria in the ileum participate in the promotion of RV infection. Mixed infections of children with RV and *enteropathogenic E. coli* resulted in an increase in the disease severity score compared to infection with RV alone [30].

## 3. Intestinal Microbiome and Immune Responses

### 3.1. In Humans

From observational studies, it has been recognized that the immune responses of children to oral or parenteral vaccines in low-income countries are often weak and that this finding correlated with the particular composition of the gut bacteriome of these children [31]. The presence of *Clostridia* and *Proteobacteria* correlated with a favorable immune response to rotavirus vaccination in Pakistan [32]. In Ghana, enrichment of *Bifidobacteria* was associated with a favorable immune response of children and that of *Enterobacteria* and *Pseudomonas* with low immune responses and lower protection [33]. The composition of the microbiome of high responders was similar to that of high responders in high-income countries [32,33]. No such differences were found in children receiving rotavirus vaccination in Nicaragua [34]. However, most evidence indicates that the composition of the intestinal microbiome is important for the improvement of vaccine efficacies [24,35]. The reasons for decreases in immune responses are complex. Besides the composition of the gut microbiome, malnutrition (including zinc deficiency and vitamin A and D avitaminoses), intestinal and extraintestinal coinfections, immunological immaturity (often linked with premature birth), the presence of maternal antibodies (transmitted via placenta), and host genetic factors play a role [3,36,37].

### 3.2. In Animals

The influence of the gut microbiome on enteric infections has been extensively studied in animal models. Thus, gnotobiotic (gn) piglets transplanted with ‘healthy’ human gut microbiota from children (HHGM: *Proteobacteria, Bacteriodetes*) or with microbiota from children with weak (‘unhealthy’) immune responses (UHGM: *Proteobacteria* and *Firmicutes*) differed in their reaction to challenge with human rotavirus: the HHGM-transplanted animals expanded *Bacteriodetes* and had less severe diarrhea and virus shedding than UHGM-transplanted animals, which maintained the high prevalence of *Firmicutes* spp. [38]. Similarly, neonatally GF piglets transplanted with a human infant’s fecal microbiome (HIFM) upon challenge had less severe rotavirus disease than nontransplanted piglets; a protein-deficient diet increased the severity of RV disease also in the HIFM-transplanted piglets [39].

In both human and animal vaccine studies, it has been shown that the composition of the gut microbiome is highly important for the efficacy of vaccines, e.g., against RV, poliovirus, and bacteria, such as *Salmonella* and *Shigella* [6,40].

## 4. Intestinal Microbiome–Host Interaction via Metabolites

In animal studies, it has become apparent that differences in the metabolism of bacteria may be important for the host’s health. Bacterial metabolites produced in the gut may enter the host via hematogenic spread and either be toxic or support the health of the gut or extraintestinal tissues. Metabolites of both microbes and host form a complex system, and humans have been termed as ‘holobionts’ in this concept [9]. Experiments aiming at discovering causal microbiome–host relationships were initiated [7]. Numerous microbial metabolites were shown to affect the host’s metabolic pathways and to be positively or negatively associated with metabolic diseases such as type 2 diabetes (T2D) or obesity [7]. For the microbiome–host relationship, the supply of polymeric compounds for bacterial fermentation in particular diets plays a very important role [8]. Some products of bacterial fermentation and selected bacterial species producing them are listed in Table 1. Of those metabolites:Short-chain fatty acids (SCFAs) have anti-inflammatory activity [41] and can act as adjuvants to vaccines [42]; butyrate-producing bacteria were found to be beneficial as probiotics (see below) in children with idiopathic nephrotic syndrome by boosting the synthesis of Treg cells [43];Products of tryptophan metabolism (kynurenine, indoles, and tryptamine) may be involved in neurological disease [44,45] or protect from colitis [44,46,47];Spore-forming gut bacteria can modulate the production of serotonin (5-hydroxytryptamine) in entero-chromaffine cells and thus affect gut motility, platelet, and CNS functions [48];Products of histidine fermentation may cause immune pathologies and be involved in asthma pathogenesis [49,50];Imidazole propionate production is correlated with an increased risk for the development of T2D [51,52];Dopamine is generated by bacterial decarboxylases from levodopa, used for the treatment of Parkinson’s disease [53,54];P-cresol, a product of tyrosine fermentation, can reduce allergic airways inflammation [55];Dietary and bacterially produced polyphenols have anti-inflammatory effects [56];Host bile acids are deconjugated and transformed into secondary bile acids in the colon, where they can inhibit the growth of *Clostridioides difficile* [57], but are also associated with an increased risk of obesity development [58];Trimethylamine-N-oxide, derived from bacterial metabolization of choline and carnitine, has been found to be associated with a risk to develop atherosclerosis and T2D [59,60,61];Sphingolipids derived from *Bacteroides* spp. metabolism are important for intestinal bacterial homeostasis [62,63].

The interplay between diet, gut microbiota fermentation, and host cellular pathways leads to a complex microbe–host-produced spectrum of metabolites, strongly suggesting that the gut microbiome affects the host’s health in a much more general way than just by its influence on immune responses.

## 5. Intestinal Microbiome and Diet

In studying causal relationships between the gut microbiome composition and the mammalian host’s health phenotype, the transplantation of human gut microbiome to experimental animals has encountered the problem that not all human bacteria may survive in the new host, also due to the fact that animal and human diets differ substantially. This has been clearly demonstrated in a study by Rodriguez et al. [67], who showed that the basal diet of mice determined the long term composition of their gut microbiome and the mouse phenotypes to a greater extent than the transfer of largely different fecal microbiomes obtained from lean or obese human donors.

In order to overcome some of these problems, a mouse model for the study of diet–microbe–host interactions has been developed using the following procedure [68]:A human simplified intestinal microbiota (SIM) consisting of 10 human bacterial strains able to metabolize dietary fibers was constructed;SIM bacteria were transferred to GF mice;Mice were kept on three different diets: chow (fiber-rich), high fat–high sucrose (low in fiber), and zero fat–high sucrose (low in fiber).

The system was used to study how the different diets may affect the abundance and the transcriptome of SIM bacteria, how SIM–diet interactions may affect the circulation of metabolites, and how this may affect the metabolism of the host. Preliminary results showed that:The diet affected the SIM bacteria colonization and their fermentation capacity;Diet–SIM bacteria interaction affected the systemic entry of SIM metabolites into the plasma of the host;The host metabolism in turn depended on the diet taken.

A microbiota-directed complementary food prototype was developed for 12–18 m old malnourished children and been found to be beneficial for weight and height gain, increase in plasma protein levels, and population by *Faecalibacterium* and *Bifidobacterium* spp. [69].

## 6. Intestinal Microbiome and Infectious and Non-Infectious Diseases

Dysbiotic microbiomes can lead to intestinal infectious diseases such as inflammatory bowel disease, necrotizing enterocolitis, irritable bowel syndrome, chronic *Clostridioides difficile* diarrhea, and extraintestinal infectious diseases [70]. Microbiota research should focus on discovering causal links between human microbiota and infectious and immune-mediated diseases [71].

A combination of dietary conditions and altered gut microbiomes was found to be associated with T2D [52,67,72], obesity [73], nonalcoholic fatty liver disease [74,75], idiopathic nephrotic syndrome [43], and cardiovascular diseases such as hypertension [76] or atherosclerotic disease [60,61,77]. In detail, the pathogenic mechanisms are complex and often systemic. The metabolic potential of gut microbiota (see above) may generate bioactive compounds, which can interact with the host in various ways [61].

## 7. Diet, Prebiotics, and Intestinal Microbes as Probiotics

Prebiotics are components of food which support the growth of gut microorganisms beneficial for human health. They mainly consist of fibers, which nondigestible in the mammalian small intestine but suitable as substrates for bacteria in the colon, mainly *Bifidobacteria* and *Lactobacillus*. A major metabolic product of bacterial fermentation of starches is SCFAs, which have antibacterial activity [64,65]. Dietary polyphenols have been shown to have anti-inflammatory and possibly prebiotic activities [56]. Gnotobiotic mice colonized by a consortium of human gut-derived bacteria were fed different food-derived fibers; by administering retrievable artificial food particles, it was possible to identify bacterial species specialized in the degradation of particular types of fiber [78]. Analysis of the microbiota of a healthy Bangladeshi birth cohort enabled the identification of several covarying bacteria—potentially leading the way toward gut microbiota repair in undernourished children [79].

Different bacteria (*Lactobacillus* and *Bifidobacterium* spp.) living in mammalian guts as commensals have been proven to act as probiotics by improving immune responses to rotavirus vaccines in gn piglets [39,80,81,82,83,84,85,86]. Probiotic bacteria have also ameliorated health in weaned piglets challenged with *Salmonella typhimurium* [87] and in children with *Salmonella* and rotavirus gastroenteritis [88,89,90]. *Lactobacillus reuteri* was shown to decrease the pathogenicity of *Clostridioides difficile* by the generation of reactive oxygen species [91].

Symbiotics are defined as a mixture of beneficial bacteria (probiotics) and fibers (prebiotics) on which the bacteria feed. They can be used as food supplements, typically consisting of lactic acid bacteria (*Lactobacillus paracasei, L. plantarum*) and plant fibers (pectin from citrus fruits, inulin from chicory root, starch from corn) and are applied as a remedy for microbe-associated acute infantile and noninfectious chronic diarrheas [92].

## 8. Intestinal Microbes as Therapeutics

Fecal microbiota transplantation (FMT) is an established procedure to treat chronic, AB-resistant, *Clostridioides difficile*-associated chronic diarrhea [93,94]. However, the overall rate of clinical cure is variable, and major adverse clinical events are not rare [7,95]. Careful risk assessment is indicated [96].

## 9. Outlook and Future Research

Knowledge of the gut microbiome, including its establishment, evolution, changes, and association with intestinal and extraintestinal diseases has enormously increased during the past 10 years. In particular, the relationship between particular gut microbiome compositions and immune responsiveness to invading microbes or vaccines has been analyzed. However, increasingly, observational studies aiming at cotemporal correlations of microbiome composition and clinical symptoms have given way to investigations in which causal relationships between gut microbiome composition, diet, complex metabolic end products (of both the microbiome and the mammalian host), and clinical phenotypes are being explored. Such data will form a rational basis for the use of gut microbiomes as therapeutic or probiotic agents.

A list of topics that remain open for future research has been collated in Table 2. The molecular biology of the interaction of gut microbiome and host, the dependence of microbiota on diet, and the influence of the joint microbiome–host metabolome on health and disease require more detailed understanding and study. Factors determining eubiosis/dysbiosis in the microbiome–host relationship have to be identified in relation to host clinical phenotypes, particularly in children from low-income countries. Mechanisms determining the influence of probiotic bacterial metabolites on the host’s immune responses have to be defined. The use of probiotic bacteria for the improvement of extended programs of immunization (EPI) in low- and middle-income countries has to be explored and optimized.

## Figures and Tables

**Table 1 viruses-13-01601-t001:** * Gut microbe fermentation products of carbohydrates, proteins, and dietary polyphenols.

Metabolite	Pathway	Bacterial Species (Selected)	Effects	References
Acetate,	Starch and amino acid fermentation	*Bifidobacterium* spp.	Anti-inflammatory	[7,43,64,65]
propionate,		*Bacteroides* spp.	Stronger immune responses	
succinate,		*Coprococcus* spp.		
butyrate				
Short-chain fatty acids		*Campylobacter* spp.	Adjuvant for cholera vaccine	[42]
		*Clostridium* spp.		
		*Eubacterium* spp.		
Kynurenine	Tryptophan fermentation	*Fusobacterium* spp.	Neurological disorder	[44,45]
		*Pseudomonas* spp.		
Indoles		*Bacteroides* spp.	Protection from colitis	[47]
		*E. coli*		[44,46]
Tryptamine		*Clostridium sporogenes*	Treatment of migraine	[45]
Serotonin	Induction of host	*Clostridium* spp.	Gut motility	[48]
			Platelet functions	
Histamine	Histidine fermentation	*E. coli, Lactobacillus*	Immunpathology	[49,50]
		*Lactobacillus*	Asthma	
Imidazole propionate		*Lactobacillus* spp.	Risk of T2D **	[51,52]
		*Streptococcus* spp.		
Dopamine	DOPA	*Enterococcus*	Treatment of Parkinson’s disease	[53,54]
	metabolism	*Helicobacter*		
P-cresol	Tyrosine and phenylalanine fermentation	*Clostridium* spp.	Reduction of airways inflammation	[55]

Polyphenols	Dietary and bacterial	*Various* spp.	Anti-inflammatory	[56]
Bile acids	Secondary	*Various* spp.	Risk of obesity	[57,58]
			Protection against Cl. diff.	
Trimethylamine-N oxide	Choline metabolism	*Various* spp.	Risk of atherosclerosis	[59,60,61]
			Risk of T2D	

Sphingolipids	Lipid metabolism	*Bacteroides* spp.	Maintenance of gut homeostasis	[62,63]

* Adapted from [7,8,66]. ** T2D: type 2 diabetes.

**Table 2 viruses-13-01601-t002:** Topics of future research on gut microbiota.

***Molecular Biology***
Interrelationship of host and gut microbiota metabolism and influence of nutrition
Identification of metabolic pathways of gut microbiota determining strong acquired immune responses
Optimization of nutrition to favor the replication of microbiota considered relevant for strong immune responses and general health promotion
Influence of joint microbiome–host metabolome on health and disease
***Pathophysiology***
Factors determining eubiotic homeostasis and the development of gut microbial dysbiosis
Relationship of defined dysbioses with clinical phenotypes of hosts
Identification of conditions in low-income countries affecting an unfavorablecomposition of the gut microbiome
***Effect of probiotics on immune responses***
Identification of metabolites of microbes used as probiotics favoring the development of strong immune responses
Reliability of animal models for the development of human probiotics
***Optimization of microbiome in human extended programs of immunization***
Identification of probiotics for use in childhood vaccination programs in low- and middle-income countries
Identification of gut microbes universally correlated with optimal immune responses, and of others correlated with insufficient immune responses
Dependence of probiotics on the underlying microbiome composition in infants in countries of different socioeconomic standards

## Data Availability

Not applicable.

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
