# Peer review of "Significance of the Gut Microbiome for Viral Diarrheal and Extra-Intestinal Diseases"

_viruses, 2021, doi:10.3390/v13081601_

Round 1
Reviewer 1 Report
This is a very interesting and coherent report on gut microbiota and gastrointestinal infections.
There are a few questions that need to be answered.
- The intestinal microbiome
It should be noted that the mode of delivery and feeding type have a strong influence on intestinal bacteria in early infancy.
- Intestinal microbiome and immune responses
4.1. A. In humans
It should be mentioned why differences in gut microbiota can lead to weaker immune responses.
Are there any findings regarding butyrate-producing bacteria and Treg?
- Diet and prebiotics
- Intestinal microbes as probiotics
There are descriptions of prebiotics and probiotics, but it would be better to mention synbiotics as well
Reviewer 2 Report
General comments:
The author described the importance of the gut microbiome in improving the immune responses for viral diarrheal, and the host’s diet can alter the composition of the gut microbiome. The work presented for review is valuable, but I recommend to rearrange the order of paragraphs for the reader’s understanding.
Specific points:
In my opinion, improvements can be made in the introduction with a clear statement that the author want to get across to the readers.
Each paragraph is numbered. How about combining some of them? e.g. #2. The intestinal microbiome and #3. Bacteria and viruses. Also, #7+#8. And #9-11. Actually, in my opinion, the positions of #7 and #8 are weird, so I think #7 should go to line 52 and #8 should be added to #6. Or not, the #7 and 8 regarding the microbiome and disease parts need improvement.
In my opinion, the details of Kovatcheva-Datchary’s study (lines 171-187) are not necessary and should be further summarized.
For greater completeness, reviewing the gut microbiota alteration by intestinal disease is necessary (https://doi.org/10.1038/s41598-018-37162-w, https://doi.org/10.1080/19490976.2020.1754714). The composition of gut microbiota to increase the immune response can be included (https://doi.org/10.1016/j.vaccine.2018.04.091).
Reference
MDPI’s reference style: In the text, reference numbers should be placed in square brackets [ ], eg. [1]. In the table 1, too.
In addition, No dot between journal name and year, No date in the reference style.
- Author 1, A.B.; Author 2, C.D. Title of the article. Abbreviated Journal Name Year, Volume, page range.
Please review your manuscript for TYPOS and grammatical errors.
L16: Clostridium difficile: italicized
L17: resistent-> resistant
L82, L101: Remove “A” and “B”
L83, L114: Add “,”
L98: linekd->linked
L108: firmicutis->firmicutes
L109, L176: germfree (GF) -> GF
L127: Add “]” after 2016
L129: Indol->indoles; tryptamin-> tryptamine?? In Table 1, too?
Table 1: spp-> unitalicized; butirate->butyrate; Short chain fatty acid-> Short chain fatty acids or SCFAs; on line of p-cresol, sp->spp; remove “i” after “inflammation”
L190: by-> unitalicized;
Table 2: unify fonts
